# Nesfatin-1 in Human Milk and Its Association with Infant Anthropometry

**DOI:** 10.3390/nu15010176

**Published:** 2022-12-30

**Authors:** Karina D. Honoré, Signe Bruun, Lotte N. Jacobsen, Magnus Domellöf, Kim F. Michaelsen, Steffen Husby, Gitte Zachariassen

**Affiliations:** 1Department of Clinical Research, Faculty of Health Science, University of Southern Denmark, DK-5000 Odense, Denmark; 2Hans Christian Andersen Children’s Hospital, Odense University Hospital, DK-5000 Odense, Denmark; 3Department of Anesthesiology, Odense University Hospital, DK-5000 Odense, Denmark; 4Strategic Business Unit Pediatric, Arla Foods Ingredients Group P/S, DK-8260 Viby, Denmark; 5Department of Clinical Sciences, Pediatrics, Umeå University, SE-901 87 Umeå, Sweden; 6Department of Nutrition, Exercise and Sports, University of Copenhagen, DK-1958 Frederiksberg, Denmark

**Keywords:** human milk components, nesfatin-1, appetite regulation, infant anthropometry, obesity

## Abstract

Breastfed infants have different growth patterns to formula-fed infants and are less likely to develop obesity later in life. Nesfatin-1 is an anorexigenic adipokine that was discovered in human milk more than a decade ago, and its role in infant appetite regulation is not clear. Our aim was to describe nesfatin-1 levels in human milk collected 3–4 months postpartum, associations with infant anthropometry, and factors (maternal pre-pregnancy body mass index (mBMI), high weight gain during pregnancy, milk fat, and energy content) possibly influencing nesfatin-1 levels. We hypothesized that nesfatin-1 levels in mother’s milk would differ for infants that were large (high weight-for-age Z-score (WAZ)) or small (low WAZ) at the time of milk sample collection. We used enzyme-linked immunosorbent assay to detect the nesfatin-1 concentration in milk samples from mothers to high WAZ (*n* = 50) and low WAZ (*n* = 50) infants. We investigated associations between nesfatin-1 levels and infant anthropometry at 3–4 months of age and growth since birth, using linear regression adjusted for mBMI, birth weight, infant sex, and exclusivity of breastfeeding. We found no difference in nesfatin-1 levels between the two groups and no association with infant anthropometry, even after adjusting for potential confounders. However, high nesfatin-1 levels were correlated with low mBMI. Future research should investigate serum nesfatin-1 level in both mothers, infants and associations with growth in breastfed children.

## 1. Introduction

Obesity in childhood and later in life is a great challenge to public health, and the prevalence continues to increase [1]. The hypothesis of nutritional programming suggests that nutritional signals in the pre- and postnatal period may influence metabolic regulation and predispose to metabolic diseases and adiposity [2,3]. Breastfeeding affects growth in early life, and most studies agree on its protective effect against obesity later in life [4,5] although the mechanisms are not completely understood. Several authors have described the levels of different appetite-regulating hormones in human milk, known as adipokines [6,7,8]. Some of these adipokines exert anorexigenic effects, and some have obesogenic effects. However, only a few adipokines described in human milk have been associated with infant anthropometry [3] and literature on how nesfatin-1 is associated with infant anthropometry and involved in infant appetite regulation is sparse. 

Nesfatin-1 is an 82-amino acid peptide that is primarily secreted by adipose tissue and the central and peripheral nervous system [9]. Nesfatin-1 was first identified in the rat hypothalamus in 2006 by Oh I S et al., who demonstrated that injection with nesfatin-1 in rat cerebrospinal fluid decreased their food intake corresponding to dose [10]. Subsequent studies extended the distribution to numerous other brain nuclei in hypothalamus, midbrain, pons, and medulla oblongata [11,12,13] and in addition, nesfatin-1 was detected in gastric mucosa in even higher concentrations [14] Serum levels in humans were described a few years later [15,16] and in 2010 Aydin first described nesfatin-1 levels in human milk [17].

Evidence suggests that nesfatin-1 is a pleotropic polypeptide that exerts both central and peripheral effects and has a well-established anorexigenic effect through reducing food intake, increasing energy expenditure (by increasing thermogenesis, involving glucose homeostasis), and contributing to inhibition of gastric motility and emptying via the autonomic nervous system [13,18]. Nesfatin-1 from maternal milk may be an important satiety factor in breastfed infants that influences growth and energy regulation and thereby reduces the risk of childhood obesity. Despite its role in appetite regulation only one recent study, have investigated the association between nesfatin-1 levels in human milk and infant growth [19]. 

The aim of the present study was to investigate the possible role of nesfatin-1 in infant appetite regulation. We describe nesfatin-1 levels in human milk samples from lactating mothers 3–4 months postpartum and its associations with infant anthropometry at 3–4 months of age. We hypothesized that nesfatin-1 levels in milk would differ between mothers to infants with a high weight-for-age Z-score (WAZ) and mothers to infants with a low WAZ at the time of milk sample collection. Further we investigated factors (maternal pre-pregnancy body mass index (mBMI), high weight gain during pregnancy, milk fat, and energy content) possibly influencing nesfatin-1 levels.

## 2. Materials and Methods

### 2.1. Participants

The Odense Child Cohort (OCC) is a prospective birth cohort consisting of children born in the Danish municipality of Odense (190,000 inhabitants). Pregnant women were included before 16 weeks of gestation from June 2010 to October 2013. The only exclusion criterion was emigration from the municipality of Odense before giving birth. The participants answered several questionnaires, attended physical examinations, and biological material (blood samples, and milk samples etc.) was collected throughout pregnancy, infancy, childhood, and adolescence. Information on maternal age at child’s birth, maternal pre-pregnancy body mass index (mBMI), birth weight (BW), gestational age (GA), infant sex, and post-delivery parity was obtained from hospital records. From the 2874 mothers included in the cohort, dropouts left a total of 2500 families participating with 2549 children [20]. Of these, 2308 infants were seen at the first physical examination at 3–4 months of age. 

In the present study, we included 100 singleton infants whose mothers had delivered a milk sample at 3–4 months postpartum and who were among the 50 infants with the highest WAZ or the 50 infants with the lowest WAZ, Figure 1.

### 2.2. Infants’ Anthropometry

Anthropometry data were collected due to a standardized protocol by trained scientific laboratory technologists at each visit, starting from 3–4 months, described in details elsewhere [21]. In addition, two examiners determined the inter-observer agreement measured in 13 children in 2012 and 7 children in 2013. Birth weight-for-gestational age Z-scores (BWZ) were calculated using Maršal [22]. Z-score at time of milk collection was calculated according to the WHO multicenter growth reference study conducted in 1997–2003 [23]. Z-score is the difference between the measured growth of the child and the expected reference growth. WAZ, height-for-age Z-scores (HAZ), and body mass index(BMI)-for-age Z-scores (BMIZ) were calculated based on the 2006 WHO standards using the STATA module, zscore06 [24].

### 2.3. Human Milk

Human milk samples were collected between May 2012 and February 2014, when the mother’s child was seen for the first physical examination at 3–4 months of age. A sample of 30 mL was requested, but less was accepted. Of the 1150 singleton mothers who attended the 3–4-month physical examination in that period, 318 (27.6%) delivered a sample with sufficient material for further analyses. The milk samples were collected between 8 am and 5 pm, and there were no requirements regarding foremilk or hindmilk, nor for complete expression or emptying of the breast. Use of a breast pump or manual expression was optional, and the time since last feeding was not recorded.

The milk samples were stored at 5 °C. If the sample was at least 10 mL, macronutrient analysis was performed (Miris HMA, Uppsala, Sweden) to determine fat (g/100 mL) and energy content (kcal/100 mL). Within 3 days after sample collection, the remaining milk sample was centrifuged at 3600 rpm and 21 °C for 5 min. (Eppendorf Centrifuge 5702 R, Eppendorf Corporate, Wesseling-Berzdorf, Germany). The resulting fat, skimmed, and solid fractions were manually aliquoted (3.5 mL transfer pipette, Sarstedt, Nümbrecht, Germany) into three different tubes (3.6 mL Nunc CryoTubes, Thermo Fisher Scientific, Waltham, MA, USA) and stored at −80 °C. In 2018 the frozen skimmed fractions were successful shipped from Odense, Denmark to Umeå, Sweden on dry ice. 

### 2.4. Nesfatin-1 Analyses

The human milk samples were thawed, and the nesfatin-1 level was analyzed immediately in the laboratory in Umeå, Sweden, using Enzyme-Linked Immunosorbent Assay (ELISA) kits, DouSet ELISA human Nesfatin-1/Nucleobindin-2, Cat no: DY5949, Lot: 325592 (R&D Systems Inc., Minneapolis, MN, USA) and DuoSet Ancillary Reagent kit 2, Cat no: DY008. The samples were measured in duplicate, and the average concentration was calculated from the standard curve in the ELISA software. The coefficient of variation (CV) (i.e., standard deviation (SD)/average) was calculated for the two duplicate values, and the samples were reanalyzed if CV was greater than 10%. The total concentration was the average value multiplied by the dilution. All samples were diluted 20 times, but a few samples were diluted up to 40 or 80 times before the concentrations were stable. Two control samples were included in all four plates to detect possible interplate variability. The average interplate CV for the two controls was 17.5%.

### 2.5. Ethics

The study was approved by the Regional Scientific Ethical Committee for Southern Denmark (project number S-20090130, sub-protocols 12 and 18) and the Danish Data Protection Agency (case number 12/26892). The study complied with the Declaration of Helsinki.

### 2.6. Statistical Analyses

Descriptive statistics were used to describe the participating mother-infant dyads, categorized as either low or high WAZ. Continuous variables are presented as mean ± standard deviation (SD) if normally distributed, otherwise as median with interquartile range (IQR). Normality was tested using histograms and QQ-plots. Maternal and infant baseline characteristics were compared using two-sided t-test for normally distributed continuous variables, the median test for non-normally distributed continuous variables, and Fisher’s exact test for categorical variables. 

As nesfatin-1 levels were not normally distributed, data were log-transformed for comparison of mean levels in the two groups using two-sided *t*-test. Nesfatin-1 levels are presented as mean ± SD after back transformation.

Associations between total log-transformed nesfatin-1 (log[nesfatin-1]), fat, and energy levels were calculated using pairwise correlation. All variables were normally distributed. As fat and energy content could only be determined in 57 samples due to insufficient amount of material, we only calculated the correlation between log[nesfatin-1], fat, and energy content in these 57 samples. As mBMI and weight gain during pregnancy were not normally distributed, correlations between these variables and log[nesfatin-1] levels were calculated using Spearman’s correlation.

Confounders and intermediate factors were identified from an a priori review of published evidence [9,17] and using directed acyclic graphs (Appendix A). Maternal BMI has been associated with other adipokines in human milk [9,25] and could be a causal intermediate factor on the pathway between nesfatin-1 levels and infant weight at 3–4 months of age. Infant sex, BW, and infant formula supplementation at milk sampling (IF) are related to infant anthropometry at 3–4 months of age and could also affect nesfatin-1 levels in human milk, therefore we adjusted for these predictor variables in the linear regression analysis investigating associations between log[nesfatin-1] milk levels and several anthropometric outcomes. Analyses were performed separately in the predefined variables, and the criteria to use multiple linear regression were met. Due to multiple testing in this analysis, the Bonferroni correction was used to calculate the critical value of *p*, resulting in a significance level at 0.004. Otherwise, level of significance was set at 0.05. Data were analyzed using STATA/BE version 17.0 (StataCorp, College Station, TX, USA).

## 3. Results 

### 3.1. Participants

An overview of maternal and infant characteristics for the two WAZ groups is shown in Table 1. There were no significant differences between the WAZ groups regarding maternal age, parity, educational level, smoking status, or weight gain during pregnancy. Unsurprisingly, we found a significant difference in mBMI. None of the mothers in this study had gestational diabetes (GDM). 

The infant group differed significantly in birth weight, birth length, early formula supplementation, and infant weight at the time of milk sampling. There was no difference in distribution of sex and gestational age, Table 1. A previous study by the authors found no significant differences between duration of exclusive breastfeeding (*p* = 0.105), season at milk sample collection (*p* = 0.838), or infant age at time of milk sampling (*p* = 0.735) [21].

### 3.2. Nesfatin-1 Levels in Human Milk, Low and High WAZ Group

We found high inter-individual variation in nesfatin-1 levels across all 100 milk samples, ranging from 0.96 to 87.76 ng/mL. The mean nesfatin-1 level across all milk samples was 7.25 ± 2.04 ng/mL. In the low WAZ group the mean level was 7.80 ± 2.31 ng/mL and in the high WAZ group the mean level was 6.73 ± 1.77 ng/mL. 

In the low WAZ group, we detected two outliers with very high nesfatin-1 levels (81 ng/mL and 88 ng/mL). The baseline maternal and infant characteristics of these outliers did not differ from those of the remaining 48 mother-infant dyads in the low WAZ group. The nesfatin-1 concentration in the two outlier samples differed from the other samples by having an unstable concentration, as these two samples were diluted 80 times before stable concentration. This was the case in all four plates, resulting in these outliers being excluded from further analyses. 

The nesfatin-1 levels in human milk in the two groups are shown in Figure 2. There was no difference in mean log[nesfatin-1] between low and high WAZ groups (*p* = 0.7). Inclusion of the two outliers from the low WAZ group did not change the difference significantly (*p* = 0.3).

### 3.3. Nesfatin-1 and Infant Anthropometry at 3–4 Months of Age

In the unadjusted analyses with log[nesfatin-1] as the predictor variable, we found no associations between nesfatin-1 levels and any of the anthropometric measures (Appendix A). Adjusting for the identified intermediate and covariates did not influence this finding, Table 2. Data shown in Table 2 are for the fully adjusted model including log[nesfatin-1] mBMI, infant sex, BW, and IF as predictor variables. 

### 3.4. Factors Possibly Influencing Nesfatin-1 Levels in Human Milk

We found no associations between nesfatin-1 levels and energy (*p* = 0.7) or fat content (*p* = 0.65) in the milk samples. There was no association between nesfatin-1 levels and high weight gain (>15 kg) during pregnancy. (*p* = 0.22) However, there was a significant negative correlation between high nesfatin-1 levels in human milk and low mBMI (*r* = −0.26, *p* = 0.01).

## 4. Discussion

### 4.1. Nesfatin-1 Levels in Human Milk

We believe this is the first study to investigate nesfatin-1 levels in human milk from mothers breastfeeding infants at 3–4 months of age. Infants were categorized as high or low WAZ at the time of milk sample collection. Surprisingly we found no difference in nesfatin-1 levels comparing the two groups. Selecting 50 infants with the highest and lowest WAZ at the time of milk sampling, we expected to find a difference. Reasons we did not find a difference, could be explained by a too small sample size, and our result may be obscured by the fact that not all the infants were exclusively breastfed at the time of milk sample collection, which is a limitation of the study. However, there was not a significant difference in the number of infants receiving formula supplementation (IF) in high versus low WAZ groups, Table 1. 

The mean nesfatin-1 level in the milk samples was 7.25 ng/mL, but there was wide individual variation. Three previous studies have analyzed nesfatin-1 levels in human milk [17,19,26]. These studies report a large variation in mean levels ranging from 0.27 to 450 ng/mL. Reports on nesfatin-1 levels in human milk must be interpreted with caution since results might be influenced by pasteurization methods, whether analysis were conducted on whole or skimmed milk samples, storage time before analysis, and time of day of milk sampling. These information’s are missing in one or more of the studies. All the studies [17,19,26] including the present, analyzed nesfatin-1 levels in milk samples using commercial ELISA kits, but from different companies. A 10-fold difference in ghrelin levels has been reported depending on the company producing the ELISA kits [27] and this could explain some of the variation in nesfatin-1 levels across the studies. Consequently, no reference value for nesfatin-1 levels in human milk has yet been established.

### 4.2. Nesfatin-1 Levels in Human Milk and Infant Anthropometry

Another recent study investigated associations between nesfatin-1 levels in human milk and anthropometry at 1 and 4 months of age in 20 infants born small for gestational age [19]. In line with our study, they did not find any correlations between anthropometric measurements and nesfatin-1 levels at 4 months of age, although all the infants were exclusively breastfed [19]. The findings in both studies support the complexity in infant appetite regulation and how breastfeeding protects against obesity later in life and emphasize that it is a complex interaction between several mechanisms, we still not fully understand.

### 4.3. Nesfatin-1 Levels in Umbilical Cord Blood and Birth Weight

Two studies have investigated nesfatin-1 levels in umbilical cord blood and associations with birth weight [28,29]. The first study found no correlation, but it was not adequately powered to examine neonatal outcomes. The second study found significantly lower nesfatin-1 concentrations in umbilical cord blood of large-for-gestational-age newborns compared with newborns appropriate for gestational age. 

### 4.4. Nesfatin-1 Levels in Serum and Associations with Childhood Weight

Regarding older children, five studies have explored serum nesfatin-1 levels in obese children (BMI > the 95th percentile) versus age- and sex-matched healthy children of normal weight. These results are conflicting, however [30,31,32,33,34]. One study reported significantly higher serum nesfatin-1 levels in obese children compared to normal-weight children [31] while the other studies reported significantly lower serum nesfatin-1 levels in obese children compared to normal-weight children (although one study only found this difference within girls) [33]. A sixth study reported lower serum nesfatin-1 levels in healthy children who were underweight (defined as WAZ < 2 SD) and had poor appetite compared to healthy children with normal body weight [35]. Overall, these results suggest that nesfatin-1 is involved in some way in appetite regulation and metabolism during childhood.

### 4.5. Milk Sample Handling Affects Hormone Levels in Human Milk

The main weakness of the present study is that the findings might be affected by the lack of standardized milk sample collection and the long storage time before the samples were analyzed. However, the observed significant correlation between milk nesfatin-1 concentration and maternal BMI suggests that the measured values have some validity.

The milk samples were refrigerated for three days, centrifuged by 3600 rpm for 5 min, and stored at minus 80 °C until analysis four to six years later. Evidence suggests that storage at refrigerator conditions (5°) is safe until 96 h, without affecting fatty acid and lipid composition or oxidative status [36] and Ramirez-Santana et al. concluded the same regarding bioactive immunological factors [37]. While frozen storage at −80 °C is generally considered safe until 9 months [38,39] Ramirez-Santana et al. found reduced IgA, IL-8, and TGF-β1 after 12 months of frozen storage [37]. Interestingly, Chang et al. found that leptin concentration remained relatively unchanged throughout freezing, heating, and pasteurization in contrast to lactoferrin, lysozyme and s-IgA [40]. Another study showed only 5% reduction in adiponectin, leptin, resistin, and ghrelin content after storage for 1–3 months [41]. This could be the case for other adipokines including nesfatin-1, but the effect of several years of freezing on hormones in human milk has not been investigated except for insulin, which has been reported stable for at least 2.5 years [42].

### 4.6. Factors Influencing Adipokines in Human Milk 

Regarding human milk leptin, one study reported a 50% higher concentration in whole milk compared with skimmed milk, and no differences were found between pre- and post-feed samples in either whole or skimmed milk [43]. Regarding adiponectin levels, no difference was reported between whole and skimmed milk, and there were no differences between pre- and post-feed samples [41]. We do not know if this is the case for nesfatin-1, but despite a difference in concentration between whole and skimmed milk, we would theoretically still be able to investigate the correlation between human milk nesfatin-1 levels and infant anthropometry. The fact that leptin and adiponectin levels did not differ between pre- and post-feed samples could reduce the risk of milk sampling bias in our study.

We did not find significant associations between milk nesfatin-1 levels and fat and energy content in 57 milk samples. Associations between nesfatin-1 levels and fat and energy content in human milk have not previously been investigated, although a positive correlation has been reported between milk fat content, leptin, and adiponectin levels [25].

We found that high nesfatin-1 levels in our milk samples correlated with low maternal pre-pregnancy BMI. This is not in line with previous findings of no correlation between mBMI and nesfatin-1 levels in human milk [17]. However, the study by Ayden only included 20 women, of whom 10 had gestational diabetes, and did not specify whether the correlation between maternal BMI and milk nesfatin-1 levels were colostrum or mature milk, nor when the mature milk samples were collected postpartum. Second, all 20 women were obese with a mean BMI 33.2 (gestational diabetes) and 32.0 (healthy controls), with no indication of whether this was pre-pregnancy BMI or BMI during lactation. In the present study, all women had median BMI within the normal range.

Regarding other adipokines, there is only reported a significant positive correlation between mBMI and leptin levels in human milk [9,25]. Results for adiponectin are conflicting, and no correlation between mBMI and ghrelin or resistin has been found [9].

### 4.7. Milk Expression Methods, Circadian Patterns, and Adipokine Levels

Nesfatin-1 levels may vary between foremilk and hindmilk or may have a circadian variation, but the literature is sparse. One study found that leptin and ghrelin levels differed between foremilk and hindmilk samples [44] while a recent systematic review was inconclusive on diurnal variation in leptin levels (the only adipokine investigated for a circadian pattern) [45]. The effect of breast pump versus manual expression has not been investigated for all biological components, but a study on macronutrients indicated that the two methods gave similar findings [46].

The mechanisms by which nesfatin-1 and other adipokines influence the infant appetite regulation and metabolism may be more complex than previously believed. This is reflected in the conflicting results in correlations between infant anthropometry and leptin and adiponectin levels in human milk. This emphasizes the need for standardized protocols for milk sample collections and an appropriate method for analysis. 

Finally, the effect of milk hormones depend on their bioavailability and the amount of hormone reaching the gastrointestinal tract. The present study was not designed to conclude anything on this issue. Estimating the infant’s total daily intake of a specific component rather than analyzing a correlation with a selected concentration in human milk at a specific timepoint could give more accurate and comparable results [46].

### 4.8. Nesfatin-1 Levels in Infant and Maternal Serum

It would have been desirable to have nesfatin-1 levels in infant serum at the same timepoint as the milk analyses. To our knowledge, no studies have investigated serum nesfatin-1 levels in infants, and the only study that has analyzed nesfatin-1 levels in maternal serum [17] found no significant associations between human milk and maternal serum levels. It remains unclear how nesfatin-1 is secreted into human milk, how it is absorbed from the gastrointestinal tract, and what degree of bioavailability it has. However, the correlation between high milk nesfatin-1 and low mBMI in the present study might suggest that nesfatin-1 levels in human milk reflect maternal serum levels rather than the amount secreted by adipose tissue in the breast. The receptor-mediating effects of nesfatin-1 and the signaling cascade are yet to be investigated [13]. 

A strength of the present study is the large sample size, as nesfatin-1 was analyzed in milk samples from 100 mother-infant dyads. Most studies analyzing adipokines or other bioactive components in human milk are performed in much fewer samples, often below 50 [9,17,19]. ELISA is the method preferred for analyzing adipokine concentration in human milk, and the method is well validated [9,46]. The data on infant anthropometry was collected in a standardized way and was validated by testing for inter-observer agreement.

## 5. Conclusions

Despite a significant negative correlation between low mBMI and high human milk nesfatin-1 levels, we found no association between mature human milk levels of nesfatin-1 and infants’ weight-for-age Z-score at 3–4 months of age in 100 mother-infant dyads. The levels of nesfatin-1 in our study showed a 10-fold difference compared to the levels reported in similar previous studies. Standardized approaches to sampling, handling and assay methods to improve study comparability is important in future studies. More research is needed to determine what influences appetite regulation in breastfed children. Future research should also investigate serum nesfatin-1 level in both mothers, infants and possible associations with growth in breastfed children. 

## Figures and Tables

**Figure 1 nutrients-15-00176-f001:**
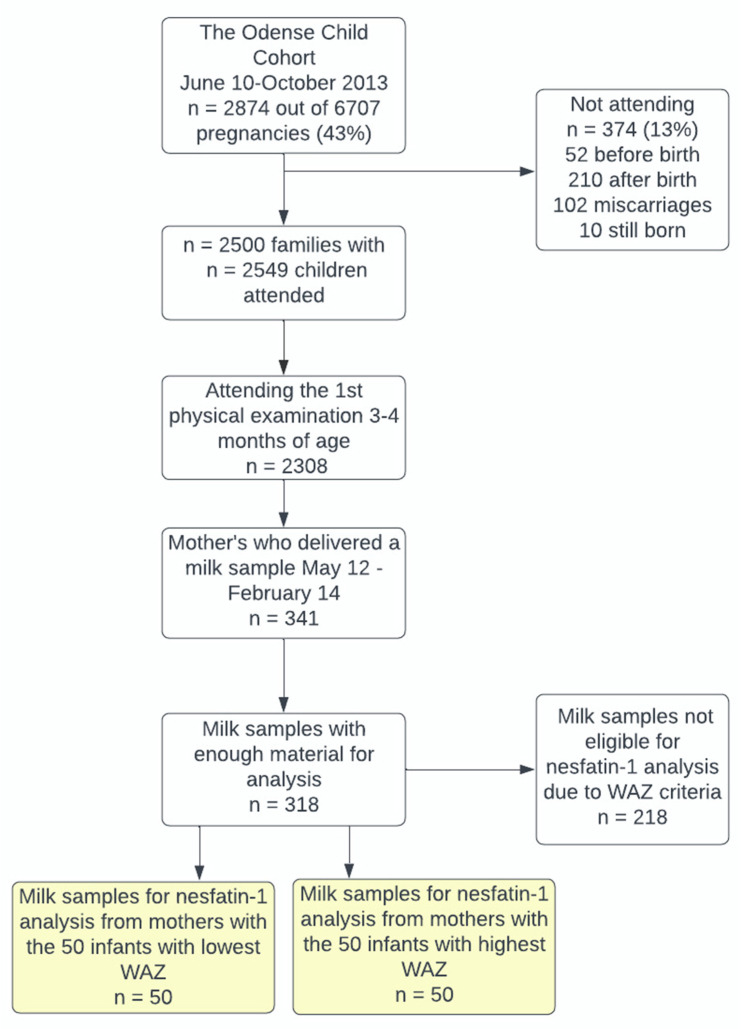
Flowchart of inclusion. (WAZ) Weight-for-age-z-score.

**Figure 2 nutrients-15-00176-f002:**
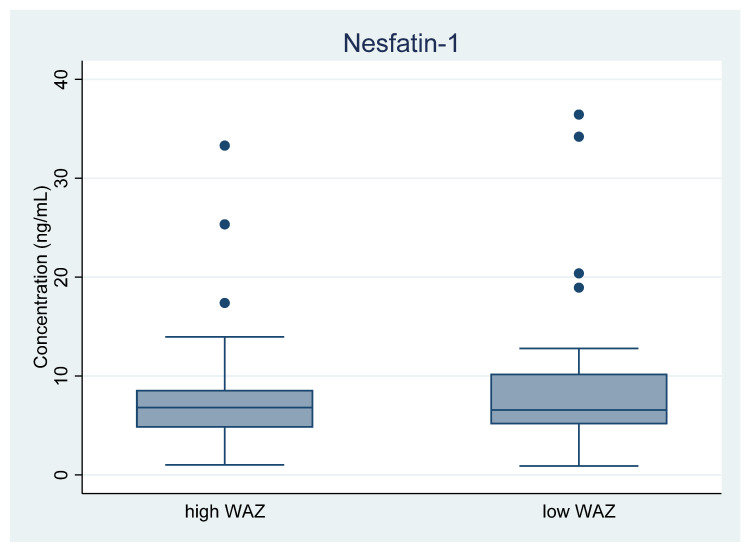
Nesfatin-1 levels in human milk, comparing high and low weight-for-age Z-score (WAZ) groups.

**Table 1 nutrients-15-00176-t001:** Characteristics of mothers and infants in the low and high weight-for-age Z-score (WAZ) groups.

	Low WAZ ^1^	High WAZ	*p*
N	50	50	
WAZ at 4 months of age, SD ^2^	−0.9 ± 0.5	1.6 ± 0.4	<0.001 *
**Maternal characteristics**			
Age at child’s birth, years	31.5 ± 5.3	30.9 ± 4.0	0.538
mBMI ^3^, kg/m^2^	22.6 (21.1–24.0)	23.5 (21.1–28.7)	0.046 *
Post-delivery parity, *n* (%)			
Primiparous	25 (50%)	22 (44%)	
Parity 2	14 (28%)	20 (40%)	0.481
≥Parity 3	11 (22%)	8 (16%)	
Educational level, *n* (%)			
Low	7 (14%)	8 (16%)	
Intermediate	21 (41%)	25 (50%)	
High	9 (18%)	5 (10%)	0.519
Unknown	13 (26%)	12 (24%)	
Smoking status, *n* (%)			
Non-smoking	50 (100%)	49 (98%)	n/a
Unknown		1 (2%)	
Weight gain during pregnancy, *n* (%)			
>15 kg	13 (26%)	20 (40%)	
≤15 kg	31 (62%)	27 (54%)	0.275
unknown	6 (12%)	3 (6%)	
Gestational diabetes, *n* (%)			
Yes	0	0	
No	50 (100%)	49 (98%)	n/a
unknown		1 (2%)	
**Infant characteristics**			
Sex, *n* (%)			
Female	23 (46%)	22 (44%)	1.00
Male	27 (54%)	28 (56%)	
Gestational age, days	279 (273–285)	285 (277–295)	0.071
Birth weight, g	3253 ± 544	3893 ± 487	<0.001 *
BWZ ^4^, SD	−0.7 ± 1.1	0.6 ± 1.1	<0.001 *
BWZ group, n (%)			
≤−2 SD	4 (8%)	0	
>−2 SD and ≤2 SD	45 (90%)	44 (88%)	0.024 *
>2 SD	1 (2%)	6 (12%)	
Birth length, cm	50.8 ± 2.5	53.3 ± 2.1	<0.001 *
Early formula supplementation, *n* (%)			
Yes	7 (14%)	15 (30%)	
No	34 (68%)	25 (50%)	0.048 *
Unknown	9 (18%)	10 (20%)	
Exclusive breastfeeding, at time of milk sampling *n* (%)			
Yes	17 (34%)	26 (52%)	
No	24 (48%)	15 (30%	
Unknown	9 (18%)	9 (18%)	0.076
Infant weight at time of sampling, g	6053 (5770–6490)	8140 (7630–8565)	<0.001 *

^1^ (WAZ) weight-for-age Z-score. ^2^ (SD) standard deviation. ^3^ (mBMI) maternal pre-pregnancy BMI. ^4^ (BWZ) birth weight-for-gestational age Z-scores. * Significant results are marked with an asterisk.

**Table 2 nutrients-15-00176-t002:** Multivariate linear regressions for anthropometrical measures and growth outcome.

Predictor Variables	Log[nesfatin-1] ^1^ ng/mL	mBMI ^2^	BW ^3^	Infant Sex	IF ^4^
** Outcome Measure **	**β (95% CI), *p***	*p*	*p*	*p*	*p*
** Δ Weight since birth (g) **	68.57 (−326.80–463.94), **0.73**	0.06	0.97	0.03	0.41
** Δ WAZ ^5^ since birth (SD) **	0.13 (−0.31–0.58), **0.51**	0.11	0.001 *	0.81	0.11
** Δ Weight since birth per day (g) **	0.49 (−2.30–3.27), **0.73**	0.12	0.87	0.05	0.03
** Length at sampling (cm) **	0.19 (−0.74–1.13), **0.68**	0.89	<0.001 *	0.16	0.28
** Weight at sampling (g) **	68.57 (−326.80–463.94), **0.73**	0.06	<0.001 *	0.03	0.41
** Abdominal circumference (cm) **	0.08 (−1.10–1.25), **0.89**	0.35	<0.001 *	0.61	0.02
** Triceps skinfold thickness (mm) **	0.17 (−0.57–0.91), **0.65**	0.10	0.98	0.71	0.33
** Subscapular skinfold thickness (mm) **	0.05 (−0.46–0.55), **0.86**	0.006	0.98	0.87	0.33
** WAZ (SD) **	0.08 (−0.33–0.50), **0.69**	0.08	<0.001 *	0.62	0.09
** HAZ ^6^ (SD) **	0.09 (−0.25–0.43), **0.61**	0.84	<0.001 *	0.01	0.99
** WHZ ^7^ (SD) **	0.05 (−0.37–0.47), **0.8**	0.01	0.03	0.33	0.02
** BMIZ ^8^ (SD) **	0.05 (−0.38–0.48), **0.81**	0.02	0.004	0.5	0.03

^1^ (Log[nesfatin]) log-transformed nesfatin-1, ^2^ (mBMI) maternal pre-pregnancy body mass index, ^3^ (BW) birth weight, ^4^ (IF) infant formula supplementation at milk sample collection, ^5^ (WAZ) weight-for-age Z-score, ^6^ (HAZ) height-for-age Z-score, ^7^ (WHZ) weight-for-height z-score, ^8^ (BMIZ) body mass index z-score. For the covariates included, only *p*-values (*p*) are shown (four last columns). The critical value of *p* at a false discovery rate of 5% was 0.004. *p*-values below this critical value are indicated with an asterisk.

## Data Availability

The data presented in this study are available on request from the corresponding author.

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
