# Peer review of "Nesfatin-1 in Human Milk and Its Association with Infant Anthropometry"

_nutrients, 2022, doi:10.3390/nu15010176_

Round 1

Reviewer 1 Report

Comments:

Section 1: Abstract:

Q1. This article is interesting, but the expressions need to be modified for further improvement including syntax errors. As a result, “We found no difference in nesfatin-1 levels between the two groups and no association with infant anthropometry, even after adjusting for potential confounders” presented no research significance. In this study, a lot of results showed differences between groups, which were suggested to be strengthened.

Section 2: Introduction:

Q2. The threats to public health in the 21st century of obesity in childhood were suggested to be enriched.

Q3: Introduction of “Nesfatin-1” in recent three years was suggested to be supplemented, including citing the references.

Q4: The last paragraph of Introduction section needed to be further improved.

Section 3: Materials and Methods:

Q5. Did the “Nesfatin-1” levels ranged with the time and eating habits of mother, and how this data was controlled?

Section 4: Results:

Q6. What did “-0.9” of WAZ at 4 months of age, SD in table 1 mean?

Q7. The results in this section were not well presented, especially the results with significant differences.

Q8. Why did not the supplemental results display in the main text?

Section 5: Discussion:

Q9. In this study, did only figure 2 involve in nesfatin-1? Other results presented no significant differences with nesfatin-1 levels, thus I would assume that the lots of expressions were not appropriate, which was suggested to be further reorganized and rearranged.

Q10. As presented, nesfatin-1 levels affect lots of indicators of body, while in this article, nesfatin-1 levels showed no significant impacts on any indices. Much more convincing explanations were needed to be supplemented.

Section 6: References:

Q11. The papers in the past three years was lacked and the reference formats were inaccurate.

Reviewer 2 Report

The authors present a very interesting paper about the possible role of nesfatin-1 from human milk in the growth of breastfed infants. I sincerely don't find any main correction to do to their product. In the attached file, though, authors can find some suggestions about possible improvement to be done, in case future studies on the same topic are planned.

Just one minor correction: "mBMI" is not disclosed both in the abstract and in the text; the first time its meaning is shown is in a table. The authors should correct this.

Author Response

Dear reviewer.

Thank you for your comments and for reading my manuscript. 

I have now disclosed "mBMI" both in the abstract and in the text the first time it appears.

Round 2

Reviewer 1 Report

Thanks for these modifications and kind responses, I have no any other comments.